# Unlocking the Entrepreneurial State of Mind for Digital Decade: SMEs and Digital Marketing

Eliza Nichifor , Radu Constantin Lixăndroiu, Cătălin Ioan Maican, Silvia Sumedrea *,
Ioana Bianca Chițu, Alina Simona Tecău and Gabriel Brătucu

Faculty of Economic Sciences and Business Administration, Transilvania University of Brașov, Colina
Universității Street No. 1, Building A, 500068 Brașov, Romania
* Correspondence: silvia.sumedrea@unitbv.ro

**Abstract:** Given the current technological changes, entrepreneurs need to transform their businesses and become more and more digitalized. The process of digital transformation through digital marketing tool adoption is decisive for SMEs' economic development. It may lead to an increase in revenues, may contribute to the transformation of business models, and may contribute to the increase in the sustainability of SMEs. Aspiring to understand what entrepreneurs think and perceive about their digital knowledge and tools to promote their business online, the authors performed a study to determine and quantify the perceived usefulness of digital tools used by SME owners, by using two different methods of research. They are represented by quantitative analysis and multi-criteria decision analysis, used to analyse the motivations, intentions, and characteristics of 333 entrepreneurs and the businesses they run. As a consequence, a matrix with 12 typologies of entrepreneurs was generated according to their behaviour towards digital marketing tools implemented for business. The obtained outcomes show the incremental potential interest of specialists and academia for SME entrepreneurs with intermediate digital knowledge. In this context, they might want to improve their skills to achieve entrepreneurial resilience with self-learning opportunities.

**Keywords:** small-business resilience; digital marketing; entrepreneurial matrix; digital knowledge

## 1. Introduction

Nowadays, digitalization is considered among the most important forces in innovation and entrepreneurship [1–3], but adopting and implementing it requires proper management of organisational, cultural, and social changes that take place in enterprises and not just passive use of digital technologies [4].

Currently, the digital development of small and medium enterprises (hereinafter SMEs) and the people involved in their activity is a priority, not only at the national but also international level [5,6]. A successful digital transformation of SMEs is decisive for economic development and societal progress, helping combat unemployment and reducing gender disparities [7–10]. Therefore, the digital development of small businesses, the ease of addressing the target audience directly (through digital communication and promotion), together with an increased market competitiveness, should be among the top priorities for today's business owners. SMEs need to follow current trends, dictated by the emergence of digital technologies, to maintain their place in various highly competitive markets [11–13] and transform their organisational culture into an innovative one based on digital technologies [14,15]. The digital transformation may lead to an increase in revenues, may contribute to the transformation of business models, and may contribute to the increase in the sustainability of SMEs [16]. Major premises in obtaining these results are based on good strategic management that can overcome digital barriers, such as invested capital, IT infrastructure, and the quality of human resources. In addition to these, consumer relations and employee behaviour are important in the management and digitalization of SMEs [17].

As a response to such complex challenges, researchers propose four levels of digital technology adoption by SMEs: "digital awareness, digital inquiry, digital collaboration and digital transformation" [18]. Factors influencing the digital transformation include both internal factors (such as resources, skills, and business models) and external factors (such as "external capabilities fit, resources fit, government regulations, and industry-related factors") [19].

For the purposes of this research, evidence from Romania with quantitative research was conducted, by analysing the motivations, intentions, and characteristics of 333 entrepreneurs and the businesses they run in the years 2020–2021. Due to the major impact that SMEs have in the global economy, researchers intend to enrich the scientific literature with a perspective on the digital tools used and the behaviour of entrepreneurs to help them choose the most adequate actions for strategies adapted to the high-demanding digital needs to aid them in building the necessary resilience frame for survival and even thriving in various competitive markets.

Based on this mission, the critical research points are represented by the following questions:

How do entrepreneurs self-assess their level of digital knowledge?

How important do they consider the online presence of the business in search engines given the present context of business survival?

What is their attitude towards digital marketing tools?

To answer these questions, the authors proposed the following objectives: (O1) to determine the level of digital knowledge of SME entrepreneurs, (O2) to measure the perceived usefulness of online presence and specific digital marketing tools by SME entrepreneurs, (O3) discover the future intentions of entrepreneurs regarding the resilient development of the SMEs in which they are involved regarding digital marketing, (O4) analyse the engagement of entrepreneur collaboration with specialists to gain support in digital presence development, and (O5) identifying the level of entrepreneurs' awareness regarding the role of search engine trends for online presence. The study revealed that most entrepreneurs express a desire to learn more about the digital tools they use for their online business presence. At the same time, the results promote the need to increase the attention of specialists and academia to SME entrepreneurs with intermediate digital knowledge to help them improve their skills and achieve entrepreneurial resilience with self-learning opportunities.

The paper contains six sections, starting with the introduction that contains background information. It is followed by the literature review and research methodology. The results unveil the research questions' answers, interpreted and related to the literature within the discussion section. The final part presents the conclusions, which highlight the study limits and the future research directions.

## 2. Literature Review

Verhoef et al., 2021 [20] presented a digital transformation that involves changes not only in information technology, but also in strategy, organisation, supply chain, and marketing. In SMEs, the use of digital tools can help create new channels for the distribution of goods and discover new ways to add value to consumers, but this depends on the sensing and learning capabilities of entrepreneurs [21].

In this respect, Chatterjee et al., 2021 [22] mentioned the impact of the perceived usefulness, ease of use, and the intention to make changes on the adoption of a new digital technology and on the customer relationship management improvement in the digitalization process of Indian SMEs.

However, different SMEs have different behaviours in terms of digitalization and some may need external help in integrating it into their development strategy [23]. A study of limited-resource SMEs showed that digital transformation can be more easily achieved with the help of digital platform service providers by improving entrepreneurial knowledge, developing social capital and organisational capacity, and training staff [24]. An equally important element for innovating and increasing the performance of SMEs in the digital transformation process is individual digital capabilities, which requires human resources

to have knowledge related to digitalization but also to develop relationships based on trust and commitment [8]. At the same time, it was found that the perception of digital transformation is different depending on the gender of those who lead SMEs, with women needing more assistance in adopting and implementing digital strategies, even if they are aware of their benefits [9]. Experts believe that even governments could get involved in supporting the digital transformation of small businesses by creating digital platforms for them, promoting digital payment methods, training digital skills, and building a digital collaboration ecosystem [25].

SMEs may face different contextual threats, for which different resilience responses (including knowledge, behaviours, skills, and processes) must be found, so SMEs should digitally improve their resilience as part of their strategy [4] and create a portfolio of resilience capabilities [26,27]. Entrepreneurial resilience is an important feature of entrepreneurship and expresses the ability to withstand time and quickly overcome obstacles [28], being connected with a higher probability of surviving and succeeding at both individual and organisational levels [28,29]. A study of SMEs in the United Kingdom divided them into four clusters of resilience: Attentive Interventionists, Light Planners, Rooted Strategists, and Reliant Neighbours, given the differences between them in terms of the location of the company, the interhuman connections, the impact of the external crisis environment, and the attitudes of entrepreneurs towards hard times [30].

Other specialists [31] analysed entrepreneurial resilience through the prism of five "pillars", namely, efficiency-based capability, adaptive capability, collaborative capability, change capability, and learning capability, to further understand how SMEs and entrepreneurs can face a crisis environment (such as the COVID-19 period). As one can see, the context of the coronavirus effects demonstrated not only the importance of the connection between citizens and businesses for online interaction [32], but also the need for a digital transformation of society, companies, and consumption patterns for development [33–35]. The pandemic period brought new challenges to SMEs, which are very vulnerable to changes due to the crisis [36]. They had to transform their business models with digital technologies and in the context of environmental factors to survive [37]. A pre-pandemic study [38] showed that the key factors influencing the business resilience of SMEs are represented by entrepreneurial characteristics (such as the enterprise owner's age, gender, and lifestyle), firm characteristics (e.g., financial capital, size, business age, and types), factors related to the business environment, and the effects of interactions between various factors and resources.

Consequently, for understanding the pandemic period, specialists propose a resilience approach with three components, namely: the resilience preconditions (e.g., SME's strengths and the active efforts made to manage recovery), the type of entrepreneur (who is/is not resilient), and the measures that can lead to its creation, both at entrepreneurial and company level [39,40].

The resilience for entrepreneurship is a complex theme and according to Korber and McNaughton, 2018 [41] there are six possible approaches to it, respectively: "resilience as traits or characteristics of entrepreneurial firms or individuals, resilience as a trigger for entrepreneurial intentions, entrepreneurial behaviour as enhancing organisational resilience, entrepreneurial firms fostering macro-level (regions, communities, economies) resilience, resilience in the context of entrepreneurial failure, and resilience as a process of recovery and transformation". The article is using the third in this list approach.

Highlighting the background of the research topic with the aspects presented above, the authors aspire to improve entrepreneurial knowledge, aiming to increase the digital performance of SMEs and enhance the owners' mindset and accomplishments of this type of business. Considering this fact, the design of the research was based on a technology acceptance model, adapted to digital marketing tools (Figure 1).

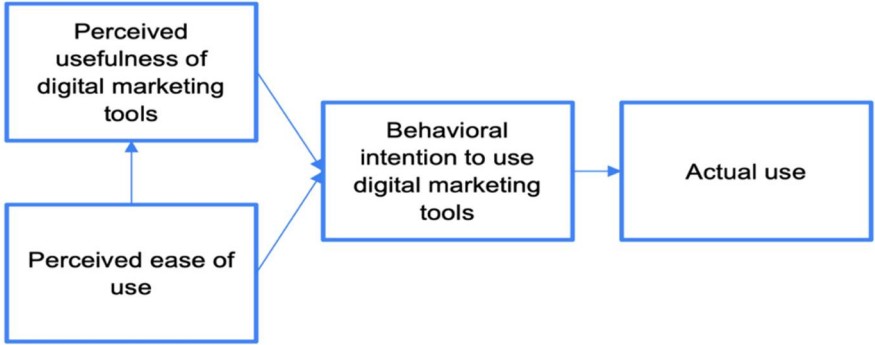

**Figure 1.** Technology acceptance model for research.

The following hypotheses were formulated:

**Hypothesis 1 (H1).** *Most Romanian entrepreneurs in the SME area are interested in specific applications of digital marketing and self-assess their level of digital knowledge as intermediate.*

**Hypothesis 2 (H2).** *Online presence and the use of tools are perceived at a high level of utility by entrepreneurs.*

**Hypothesis 3 (H3).** *Entrepreneurs perceive a direct link between resilience and digital marketing and adopt strategies for business growth with digital marketing tools.*

**Hypothesis 4 (H4).** *The collaboration of entrepreneurs with digital marketing specialists is low.*

**Hypothesis 5 (H5).** *Most respondents do not know the details of search engine trends, but if they knew them, they would use them for their business.*

### 3. Research Methodology

*3.1. Sampling and Data Collection Methods*

The study is a survey-based investigation that draws on the data collected regarding SMEs from different fields with online activity potential in Romania. The study lasted 16 months, between May 2020 and September 2021, which intentionally covers the pandemic period. The business owners targeted for the research are not only part of the population of a certain county, but the national coverage was tried for a juxtaposition as close as possible to the situation of the representative population. The researched population is represented by people over the age of 18, all entrepreneurs with an active legal form of a business.

To perform the study, it was considered to include only those companies that met the criteria for their classification in the category of micro, small and medium enterprises (SMEs). As in the international economy, in Romania, SMEs also play a critical role [42–44]. Thus, the sample included in the research belongs to a total population of almost 450,000 SMEs [45]. The European average is 58 SMEs per 1000 inhabitants and their total represents 99% of the economy, considering only the non-financial sectors (industry, trade, construction, or services).

A sample of 4500 SME representatives was selected and a process for data collection was initiated among them.

For the construction of the sample, stratified random sampling was used, the researchers having access to the database with SMEs in Romania. The studied population was divided into layers by areas of activity, then random samples were extracted from each, so as to ensure adequate sampling of all groups [46–50].

This process was performed with active online and offline campaigns aiming to attract SME owners interested in digital marketing.

Further, the selection criteria involved SME entrepreneurs that are in charge of the decision-making process on spending a marketing budget allocated to developing an online presence in the organisation. The criteria for preparing the list of digital marketing tools (used for online presence) were based on the assumption that online searches are mostly performed on the Google search engine, which led the authors to choose technologies created or used for it. Further, their spread throughout the world and the online facile access to them were considered.

Data collection was followed by checking the eligibility for each respondent. In the final sample of 377 entrepreneurs, 333 were considered eligible for data processing with SPSS software, version 28.0.1.0. The clear definition of the target profiles and the introduction of a pre-selection stage of the participants managed to adjust the sample to the relevant parameters of the population, largely avoiding the problems related to selection biases [51,52]. The philosophy of creating the survey is based on the main research question, decomposed into k dimensions, generating m questions, r explanatory factors, with s dimensions, and t indicators, to which v verification questions were added.

In this way, 20 questions determine the questionnaire, based on the following formula $N = k \cdot m + r \cdot s \cdot t + v$. In other words, the questionnaire contains 2 filter questions, 12 questions related to the organisation's performance, and 6 identification questions. It was applied by the method of personal approach through direct interviews, a great option for efficiency in behaviour studies [53]. Moreover, by selecting the direct interaction with the subjects, the authors tried to counteract several disadvantages that the CAWI (computer-assisted web interviewing) method, for example, faces [54]. Thus, a major advantage obtained by direct interviewing is represented by avoiding the impossibility of providing answers, completed by direct and permanent support to obtain the best information about the behaviour and perception of interviewees. In addition, it favoured the nexus with the represented brand. All this led to a more involved and closer discussion about the real situation of the business in which they are involved. Although the efforts related to human resources or time were higher than in any other applicable method, the authors consider that the relationship created during the interviews generated rewarding raw data for digital marketing strategy proposals presented in this paper.

*3.2. Analysis Methods*

To achieve the established objectives, the authors conducted quantitative analysis, such as descriptive analysis [55], inferential statistics [56], hypothesis testing using the t-Student [57] and McNemar tests [58], and a regression [59,60]. Furthermore, TOPSIS analysis was performed to find the ideal negative and positive cases for SME entrepreneurs [61–66]. The positive ideal case and the negative ideal case can be built with the principle of the TOPSIS method, using the variants for different criteria [66]. The distances between the two cases offer the hierarchy of the variants and is used to calculate the solution by comparing the distance relative to the positive case. From a geometric interpretation point of view, each variant is a spot in n-dimensional space ($n$ = number of criteria). Two points attract the attention in the geometric space, namely the ideal positive case and the negative ideal case, based on which the relative distances of the variants are determined.

**4. Research Results**

According to the data obtained, the sample includes most two-year-old SMEs (12%), even if the average is 8.05 years, with four employees, the most common cases being represented by companies with no employees (21.9% out of total). At the gender level, most of the SMEs included in the study are managed by males (66.2%).

Regarding the structure of the sample according to the industry, data indicate that the industries from which the most interested digital marketing entrepreneurs come are commerce and services, followed by production and technology (Figure 2).

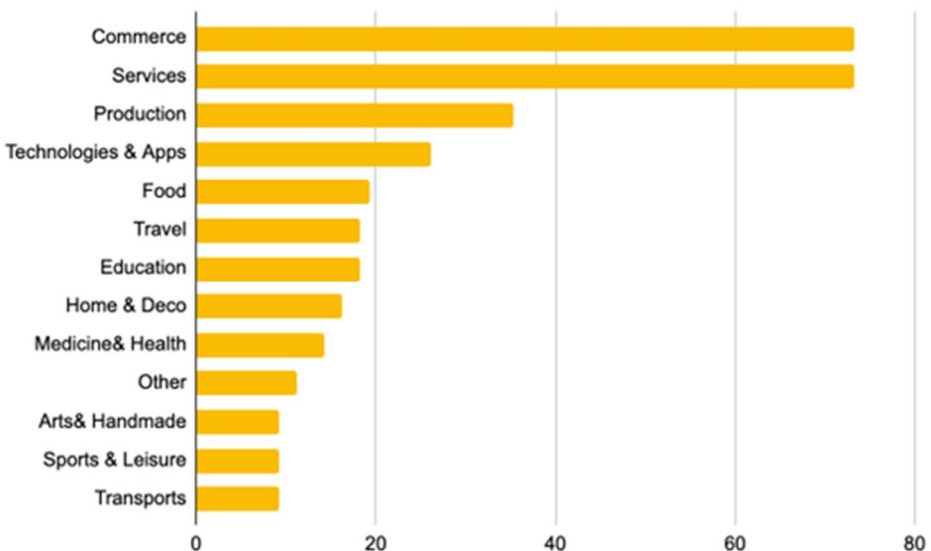

**Figure 2.** The sample structure according to the industry in which the SME operates.

The first result that caught the attention of the authors was related to the interest of the entrepreneurs granted specific digital tools to promote their business. Of the total number of participants in the study, all of them declared themselves interested in the tools selected for research. Based on the hypothesis applied in the case of the unilateral statistic test, $z_{obs} = 50$, according to the decision, the relation $z_{Obs}$ (50) $> z_{\alpha/2}$ (1.96) presents with a 95% probability that more than half of the Romanian entrepreneurs are interested in the specific applications of digital marketing.

Completing the concern for this field, objective (O1) was to quantify the self-perception of entrepreneurs regarding their level of digital knowledge. The analysis of the results generated by using the ordinal scale shows the modal value (mode = 2), which presents most of the subjects that appreciate their level of digital knowledge as intermediate (see Table 1). This fact is also confirmed by the central tendency shown by the value of the median, which reveals where the answer of the 50% subject in the sample is located.

**Table 1.** Google My Business (GMB) statistics.

|  | **How Do You Rate Your Digital Knowledge?** |
|---|---|
| Valid | 306 |
| Missing | 27 |
| Mean | 2.35 |
| Median | 2 |
| Mode | Intermediate |
| Kurtosis | −0.68 |
| Skewness | −0.24 |
| S.E. Skew | −0.14 |

The same table presents a concentration of answers related to the right symmetry, given both by S < 0 (−0.24), with the value, mean > median, but also by the value K ≠ 0 (−0.68), which demonstrates a platykurtic flattening, which means few extreme positive or negative answers.

Another relevant research question (namely $R^2$) was about the perception of the interviewed entrepreneurs related to the importance of online presence in the search engines of the business in which they are involved. For its analysis, an ordinal scale of importance was used, namely a semantic differential, where level 1 indicates the answer "Not at all important", and level 5—"Very Important".

By performing the descriptive analysis, it was obtained that of the total number of subjects, most (58.9%) consider the digital presence very important and a few consider it

almost important (7.5%). Moreover, the median shows where 50% of the sample members are located, and in this case, the subjects state that the online presence of the business is very important. The average at the level of the sample is 4.34 points, dictating the central trend of the studied sample.

To obtain a descriptive analysis as relevant as possible and close to the statistical distribution of the studied behaviours at the population level, it was desired to calculate the indicators of variation, so that there is no danger of affecting the chances of extrapolating the results to the total population. In this regard, the first step was to calculate the various indicators for the variable on the importance of the online presence of the businesses in search engines. The first indicator of variation considered is dispersion (0.875), calculating the average of the quadratic differences between the individual values of the variables and its average. Therefore, the standard deviation was calculated (0.935), followed by the standard deviation from sample means (0.051).

This confirms and demonstrates the low dispersion in the research population. Its small value is also due to the value of the standard deviation (<1) and if the sample had been larger, its value would have been even smaller.

By estimating the mean $\bar{x} = 4.33$, the result led the authors to analyse even more deeply. The descriptive analysis presents the modal and median value located at 5 points. Given that the normal distribution was used at the level of $n \geq 30$ people (z), the confidence interval was determined, $\mu \in$ (4.23 points; 4.42 points), with $E = 0.099$ and $s_{\bar{x}} = 0.051$.

Another significant result from the research (related to O2 objective) is the statistics on the digital tools used by entrepreneurs in the online activity of the company. To obtain the data, a nominal scale with the possibility of multiple choice was used. There were 307 valid answers to this question, with 778 responses. As illustrated (Figure 3), 76.3% of companies are listed online with the help of an official website, followed by the Google My Business app (GMB), which provides a presence in the search engine of the same name. The default microsite in this application is used by 33.9% of SMEs, but Google Analytics seems to be installed for 47.4% of companies. At the other end of the spectrum is consulting search engine trends, a method applied by only 6% of entrepreneurs. Further, page speed testing tools, such as Test My Site, are implemented in only 9.3% of cases.

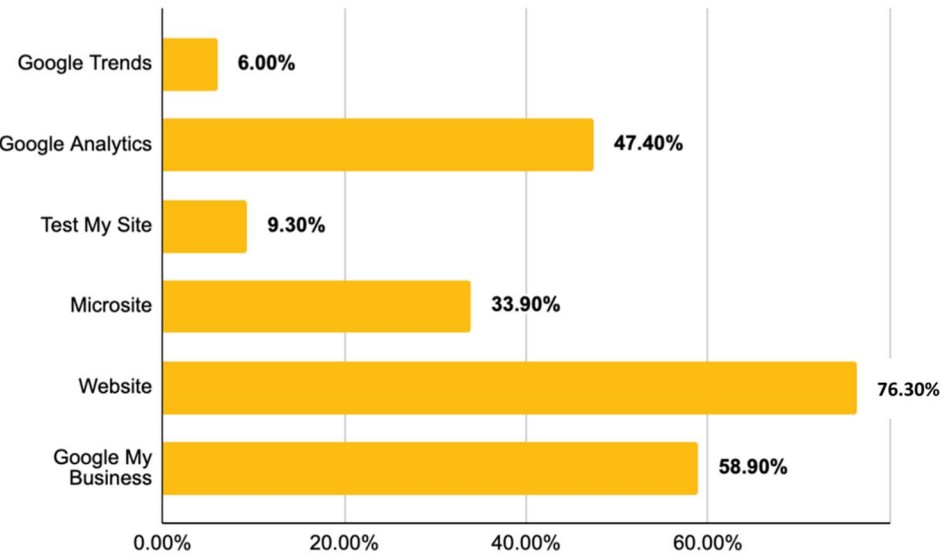

**Figure 3.** Digital tools used by entrepreneurs in the online activity of companies.

Alerts and Market Finder tools are not indicated as being used or implemented in the analysed SMEs.

The response variants of this nominal were subsequently treated as dichotomous nominal, presented below, as follows. For the first variant, descriptive statistical indicators were calculated. Based on these, the p = 58.9%, $s_p$ (0.269), and E (0.052), it can be guaranteed

with a probability of 95% that the percentage of those who use the GMB digital tool for the online activity of the company is between 53.63% and 64.17%.

According to the value obtained for $s = \sqrt{2420.79} = 0.492$, it was possible to notice the high homogeneity of the population regarding the use of this digital tool, supported by the analysis of frequencies for the answers "yes" and "no" provided by respondents, with an almost completely uniform distribution.

The standard deviation from the sample means was calculated, $s_p = 0.269$, and the authors noticed a very low value. For even greater accuracy of the calculated result, it was considered relevant to test the statistical hypothesis for the bilateral test to compare the chosen sample with the value of t, generated using the Student's test for univariate analysis. In this sense, the statistical hypotheses were:

H0: The percentage of Romanian SME entrepreneurs who use the GMB digital tool for the company's online activity is 50% ($\pi$ = 50%).

H1: The percentage of Romanian SME entrepreneurs who use the GMB digital tool for the company's online activity is different from 50% ($\pi \neq$ 50%). Running the test, the results were confirmed. The value of $t_{calc}$ = −1829.58, which does not belong to the specific range, [−1.96; +1.96] was determined, accepting the alternative hypothesis, H1. To identify possible significant possible differences between the expected and observed cases for the variable, the $\chi^2$ test was applied (H0: $O_{ij}$ = $E_{ij}$ and H1: $O_{ij} \neq E_{ij}$).

The calculation $\chi^2$ was performed and Table 2 shows the observed and expected frequencies.

**Table 2.** Observed and expected frequencies for GMB tool.

| Value | Observed N | Expected N | Residual |
|:-----:|:----------:|:----------:|:--------:|
| No | 137 | 166.5 | −29.5 |
| Yes | 196 | 166.5 | 29.5 |
| Total | 333 | | |

Comparing the value of $\chi^2_{calc}$ = 10.45 (Table 3) (also confirmed by performing the test using SPSS) with the theoretical value (calculated in Excel, with CHIINV for $\chi^2_{0.05; 1}$ =3.84) the relation $\chi^2_{calc}$ = 10.45 > $\chi^2_{0.05; 1}$ = 3.84, caused the rejection of the null hypothesis and the acceptance of H1. Based on Asymp. Sig. = 0.001, the same decision was taken. Thus, it can be guaranteed with 95% probability that there are significant differences between the observed and expected frequencies, which means that there is a link between the analysed variables.

**Table 3.** Chi-square test statistics GMB tool usage.

| | Chi-Square | df | Asymp. Sig. |
|:---:|:----------:|:--:|:-----------:|
| GMB | 10.45 | 1 | 0.001 |

Applying the same methodology for the microsite and website variables, it was obtained that, for the sample data, the percentage of Romanian entrepreneurs who use the microsite in the company's online activity is between 28.82% and 38.97%, with a standard deviation of 0.259, highlighting the potential for a link between variables, given the significant differences. Further, the use of the official website is made by Romanian entrepreneurs in proportions between 71.13% and 81.27%, with a standard deviation from the sample average of 0.741. As in the case of the microsite, the website, and the GMB, it can be guaranteed that this variable has significant differences between the observed and expected frequencies, which determine the existence of possible links regarding the analysed variable.

At the sample level, the percentage of those who use the analysis tool, Google Analytics, is 47.4%. Calculating the critical ratio, $Z_{obs} = (p - \pi_0)/s_p$ = −10.05, the relation $z_{Obs}$ (−10.05) < $Z_{\alpha/2}$ (1.96) was obtained. Hence, the percentage of Romanian entrepreneurs

who use the Test My Site application is less than 25%. In the case of Google Analytics (GA), according to the calculated estimates, the percentage of those who use GA is, in fact, between 40.35% and 54.09%, and the percentage of Romanian entrepreneurs who use the Google Trends application belongs to a range of 4.7% to 7.3%.

Desiring to know more details about the interest of SME entrepreneurs in the page-loading speed of the websites, the authors asked the subjects if they currently use any tool. In addition to obtaining the percentage of companies that have implemented such tools, the role of the question was to verify the answer given to the question related to the use of tools for digital marketing (nominal scale with multiple responses). The descriptive statistics obtained a percentage of 84.7 for respondents who use such apps, but a big difference between the answers to the variable (9.3%) and the answers to the verification question was noticed. Therefore, the authors performed a bivariate analysis, running a statistical test used on paired nominal data. In this regard, the Quinn McNemar test was applied to 2 × 2 contingency tables to identify possible changes in the answers given by the subjects. Assuming the null hypothesis that none of the two models perform better than the other, the results generated the rejection of H0 (point probability < 0.05) and acceptance of H1, which means that the marginal frequencies are not homogeneous. This result shows the undecided subjects regarding this type of digital tool.

Another interesting result related to the O2 objective was obtained using a nominal scale, with the possibility of choosing a single answer measuring the perception of SME entrepreneurs regarding the role of trends in search engines. Most of the respondents (94%) do not know the details about search engine trends, but if they did, they would use them to improve their business. On the other hand, the analysis of the answers to the question "Would you be interested in a case study for three months in which the company would benefit from support for the development of the digital presence?" shows that the most common answer is a negative one, given by 91.4% of respondents.

Considering the potential use of Market Finder as a digital tool for expanding the business internationally, the subjects were asked to say if they know how to use it to analyse foreign markets. As such, 96.7% answered in the negative. Only 2.7% of managers would know about the existence and use of digital tools for this purpose. The answers regarding the future needs of the interviewees present the fact that they thought about solving problems with Google Ads account management, optimising campaigns on Google, and analysing case studies.

A possible correlation was discovered between the intention to intensify online activity and the intermediate level of knowledge, generating a favourable framework for exploiting the development potential of 2-year-old SMEs existing on the market, in which there are no employees working in services, commerce, production, and technology. Testing the hypothesis regarding the connection between industry and the positive perception of entrepreneurs regarding the tracking intentions, it was obtained that no relation exists between them, which highlights the potential for each industry equally.

Discovering all these results (see Tables 4 and 5), the authors observed that some variables are influencing the online presence. Therefore, a multiple linear regression model with standardised regression coefficients was performed, as follows:

$$OPI = \beta_0 + \beta_1 GMB + \beta_2 WEB + \beta_3 TmS + \beta_4 GT + \beta_5 OAI + \beta_6 PSL + \beta_7 TRK + \beta_8 G + \beta_9 DK$$

where OPI is the online presence importance considered by SME entrepreneurs in search engines, GMB—Google My Business, WEB—Website, TmS—Test My Site, GT—Google Trends, OAI—Online Activity Intensification with GMB, PSL—Page speed loading analysis, TRK—Website traffic tracking, G—Gender, and DK—Digital Knowledge (self-assessed).

**Table 4.** Analysis of variance for correlation between predictors and criteria regarding the regression performed.

|  | Sum of Squares | df | Mean Square | F |
|---|---|---|---|---|
| Regression | 205.00 | 9 | 22.78 | 94.18 |
| Residual | 71.11 | 294 | 0.24 |  |
| Total | 276.10 | 303 |  |  |

**Table 5.** Coefficients for the regression.

|  | Unstandardised Coefficients | | Standardised Coefficients | | |
|---|---|---|---|---|---|
|  | B | Std. Error | Beta | t | Sig. |
| Constant | 3.54 | 0.24 | 0.00 | 14.67 | 0.000 |
| GMB | 1.49 | 0.06 | 0.78 | 23.63 | 0.000 |
| Website | 0.48 | 0.10 | 0.22 | 4.90 | 0.000 |
| TmS | −0.35 | 0.16 | −0.10 | −2.15 | 0.033 |
| GT | 0.17 | 0.14 | 0.04 | 1.25 | 0.214 |
| OAI | 0.10 | 0.06 | 0.05 | 1.65 | 0.099 |
| PLS | −0.22 | 0.14 | −0.08 | −1.64 | 0.103 |
| TRK | 0.10 | 0.04 | 0.10 | 2.32 | 0.021 |
| G | −0.3 | 0.06 | −0.01 | −0.44 | 0.662 |
| DK | −0.18 | 0.05 | −0.11 | −3.14 | 0.001 |

Examining the multiple correlation coefficient value (R = 0.86), a high correlation may be noticed between the predictor variables simultaneously with the criterion variable. The value of $R^2$ (0.74) is the correction and shows that 74% of the variation in the importance of the online presence of the business in search engines is determined by the nine β coefficients. The standard error of the estimate presents a low value (0.49) and indicates the accuracy and the reliability of the prediction model. The overall correlation between predictors and criteria is given by the ANOVA analysis (Table 4), which shows that the predictor variables correlate significantly with the criterion variable.

In other words, given the F (94.18) and the value of Sig. (0.000), the authors could reject the null hypothesis and accept that the nine predictor variables together influence the variation in the criteria. Globally, the intensification of online presence in search engines is influenced by the nine parameters but the question is whether, alone, they are significant for criteria estimation. To find a response, analysis was performed. Given the values of GT, OAI, PLS, and G (>0.05), it can be said that they are not significant for dependent variables alone. Otherwise, a positive correlation exists between the OPI and GMB, OPI and Website, OPI and TRK, and a negative correlation is between OPI and TmS and OPI and DK.

Nevertheless, the authors realised that these results can be valorised and a grouping of different entrepreneurs' typologies was performed. By mixing the variables, 12 typologies of small business owners were designed and named by the authors based on the perceived usefulness (Figure 4) and placed in the order of the complexity of digital tool use (Figure 5). All typologies were named based on the raw data collected from the point of view of digital marketing tool adoption and their perceived usefulness. According to the intersection points, each category was placed in the matrix, with the following descriptions.

The Untimely Diver typology of entrepreneurs is characterised by the subjects that do not think that Google My Business (GMB) [67] and/or microsite [68] tools are useful in business development. The same attitude about the perceived usefulness of more advanced digital tools, such as an official website and/or Google Analytics [69] and/or Test my Site [70], has the entrepreneurs named Digital Risk Takers, who adopt these digital tools. The ones who consider Alerts [71] and/or Market Finder [72] as tools that do not contribute to business development, but still adopt them, were named Digital Opportunists. The SME entrepreneurs who do not know any details about search engine trends, but if they did, they would use them to develop the business, were placed in the second column. Here, based on the adoption of each tool, three categories were created, namely, Digital Aspirant, Digitally Experienced, and Digitally Informed. The entrepreneurs who think these digital tools can be consulted but do not play a major role in developing the business were named Digital Juvenile, Digitally Savvy, and Digital Explorer, depending on which instrument they adopted. Finally, the ones who consider digital marketing tools very helpful, especially for search engine advertising,

are represented by the High-Potential runner, Digital Achiever, and Digital Investigator typologies, and included the fourth column.

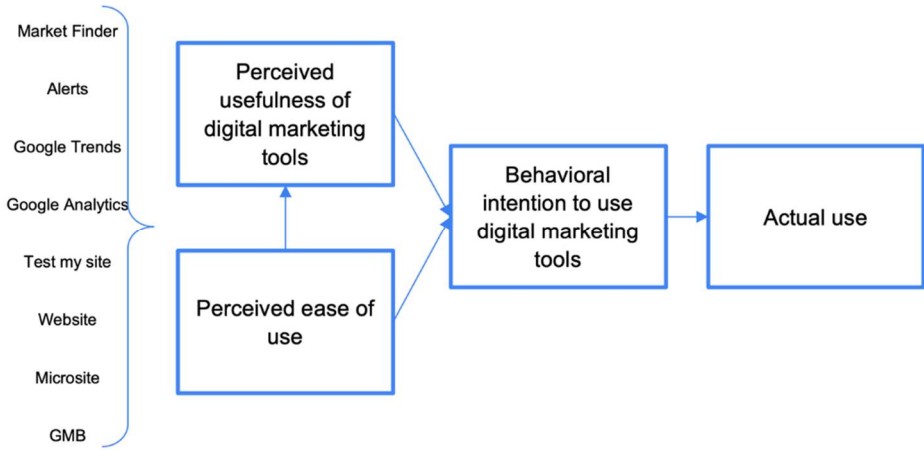

**Figure 4.** Digital tools studied and technology acceptance model.

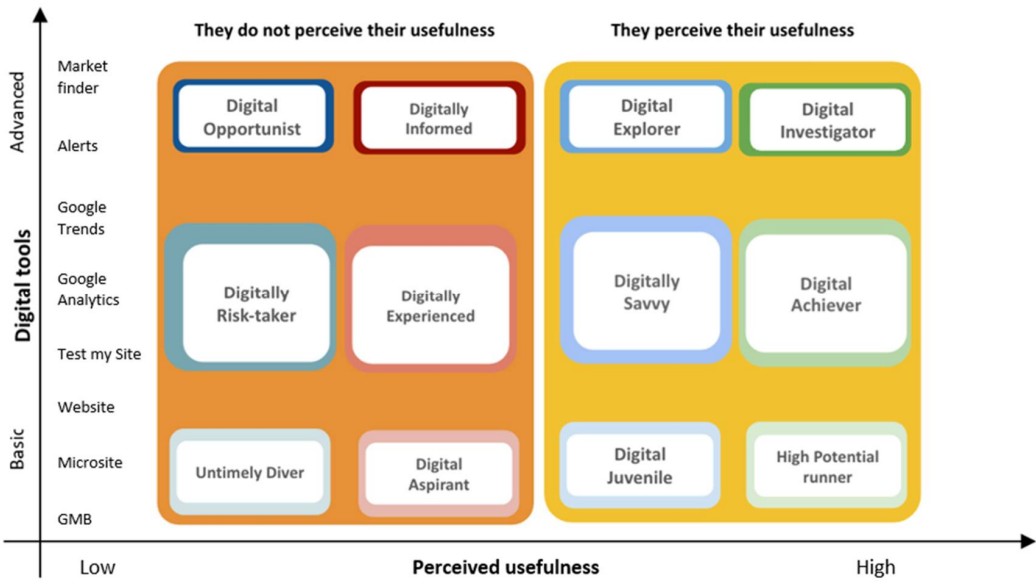

**Figure 5.** Digital Entrepreneurs matrix.

By applying the TOPSIS method, the results generated show that the positive ideal case is represented by the *Digital Investigator* typology, while the negative ideal case is the *Untimely Diver*.

According to Table 6, Digital Aspirants and Digitally Experienced entrepreneurs are the most frequent types of entrepreneurs in the sample and none of the subjects was identified as a Digital Investigator.

According to the same data, it can be observed that an entrepreneur is included in two or many typologies. This result generated the interest of the authors to approach the classification in a more complex way. The analysis allowed us to develop the framework presented in Figure 6, which shows the distribution of the entrepreneurs on points by three levels, depending on their perception about the role of search engine trends for business development, drawn up as follows:

**Table 6.** The structure of the sample by the typologies.

| Typology | Frequency |
| --- | --- |
| Digital Aspirant/Digitally Experienced | 174 |
| Digitally/Experienced | 80 |
| Digital/Aspirant | 27 |
| Digitally/Risk-taker/Digital Achiever | 11 |
| Untimely diver/High potential Runner/Digitally Risk-taker/Digital Achiever | 9 |
| Digital Aspirant/Digital Juvenile/Digitally Experienced Digitally Savvy | 2 |
| Digitally Experienced Digitally Savvy | 1 |

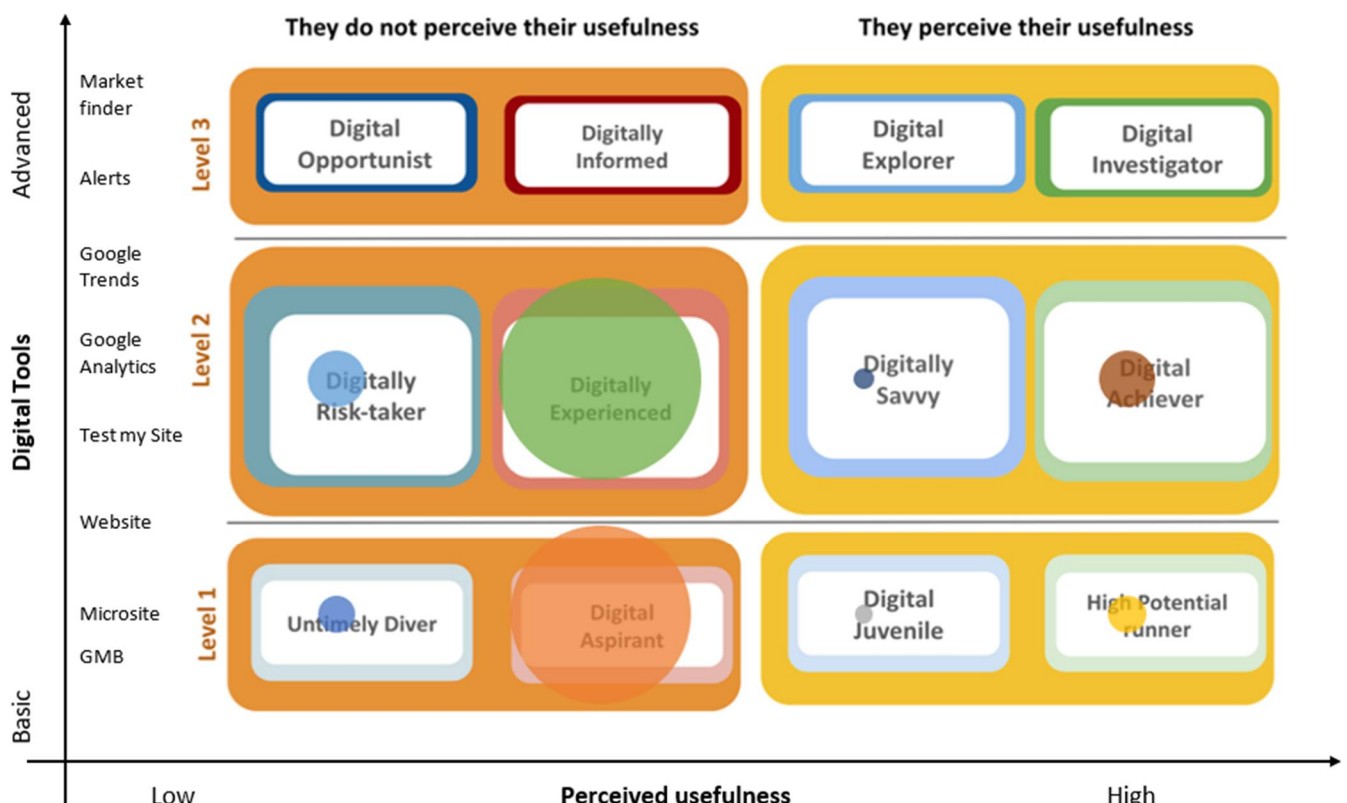

**Figure 6.** Entrepreneur frequency distribution using the three levels.

Level 1 is associated with business owners that deploy and use Google My Business features for online presence and launch the integrated microsite, a useful tool for better indexing in the Google search engine.

Level 2 is represented by those using the Website, Test my Site App, Google Analytics, and Google Trends platforms to interact, track, and analyse user behaviour.

Level 3 shows the use of different digital tools, such as Market Finder or Alerts, to be informed about detailed insights regarding business opportunities.

As is shown by the illustrated situation of the SME owners, most of them are associated with the Digitally Experienced typology, represented by 77% of participants in the study. Table 7 presents the structure of the sample, by the predominant typologies.

**Table 7.** The predominant digital entrepreneurial typologies in the sample (1—lowest perceived usefulness, 4—highest perceived usefulness).

| Complexity Level/Perceived Usefulness of Digital Tools | 1 | 2 | 3 |
|---|---|---|---|
| 1 | 3% | 61% | 1% |
| 2 | 6% | 77% | 1% |
| 3 | 0% | 0% | 0% |

## 5. Discussion

Starting from the research questions regarding the self-assessed digital knowledge level, the importance of online presence, and the attitude of entrepreneurs towards digital marketing tools, the authors achieved their proposed objectives. Further, unexpected results (Figure 7) were obtained and are presented in the next section.

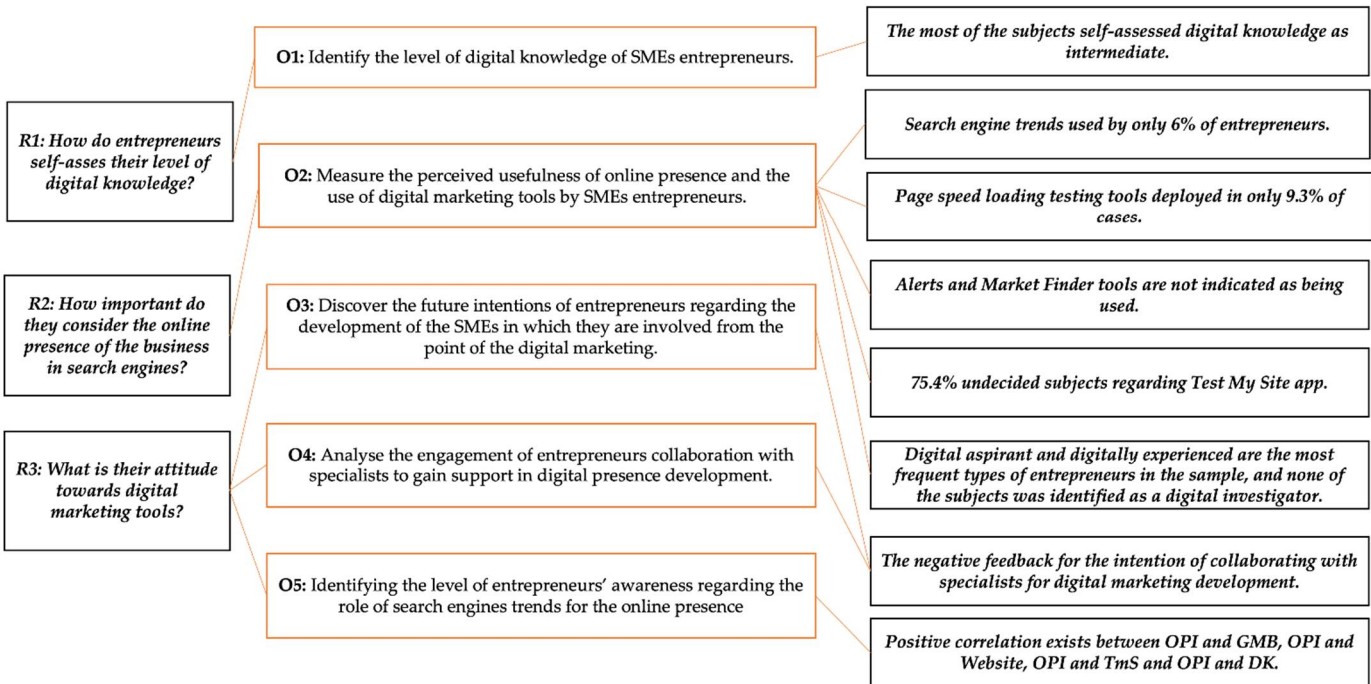

**Figure 7.** Unexpected results summary according to the research questions and objectives.

The first research question guided the authors to successfully achieve the objective related to the self-perception of entrepreneurs regarding their level of digital knowledge. It resulted in quantifying that most consider their level of digital knowledge to be intermediate and highlights one of the strengths in the research points (H1 confirmed). Secondly, another objective (O2) was formulated to determine the interest of entrepreneurs in digital marketing and conducted to the expected result of a high interest in most of them. Measuring the perceived usefulness of online presence and the use of digital marketing tools by SME entrepreneurs, researchers present most of the subjects with high interest for specific digital tools to promote their business, considering it very important (see H2). This was an expected result for the O2 objective, given the sample of the study, but the interesting fact about this research topic is represented by a notable discovery. Even SME entrepreneurs declared themselves interested in digital marketing tools and perceive the usefulness of digital presence in search engines as very important; at the same time, they do not perceive the usefulness of working for three months at least for special support to enhance digital performance and do not intend to engage in such collaboration. In other words, the researchers found an entrepreneurial state-of-mind issue here because their mindset is not aligned with the vision of changing the business model and increasing its sustainability, as Rupeika-Apoga et al. (2022) presented [16].

Specifically, the subjects assessed themselves at intermediate level in using digital marketing tools and, at the same time, they did not consider that they needed some support from specialists. This

fact is due to perceived ease of using these types of tools. Contrary to this way of thinking, most of the respondents (94%) do not know the details about search engine trends, but if they did, they would use them to improve their business, which indicates the presence of the perceived usefulness and ease of use. Further, the larger extent of use of advanced digital marketing tools (such as Google Analytics or Website) than basic ones should raise a big question mark for strategists, consultants, or academics when discussing the process of SME digital transformation. The early adoption of the advanced apps may put in danger the potential of individual digital capabilities as Scuotto et al. (2021), Chen et al. (2021), and Tarutė et al. (2018) [8,19,25] wrote about. Moreover, considering the resilience that acts as a facilitator for the relationship between entrepreneurial intent and entrepreneurial efficiency [73], building a portfolio of skills is a necessary condition in developing a new mindset. Answers to the future needs of the interviewees show the need for help with Google Ads Account Management, Google Campaign Optimization, and Case Study Analysis, but the learning or dealing method is also very important given the situation. On the other hand, the negative feedback for the engagement of entrepreneurs with specialists for support recalls the difference between SMEs in terms of digitalization, where [23] shows that SME performances are different and additional help in integrating digital tools into development strategy is needed. All these results provide the necessary information to conclude that the third and the fourth hypotheses (H3 and H4) were partially achieved (related to O3 and O4).

Moreover, the fifth goal (O5) guided researchers to relevant results for online activity in search engines, especially placing it as a real priority (so, H5 was confirmed). Intending to find a possible correlation between industry and the positive perception of entrepreneurs regarding tracking intentions, the results showed the opposite. Therefore, an equal potential for each industry exists. On the other hand, the multiple regression model applied presents the high correlation between the predictor variables simultaneously with the criterion variable. However, among the most unexpected results are some those related to this model. The negative correlation between OPI, TmS, OPI, and DK complete the scientific contribution of [9,19,21]. Particularly, these factors negatively influencing this intention could be considered to be included in the resilience portfolio capabilities [26,27] besides training for digital training [25]. Further evidence for unexpected results is presented as the Alerts and Market Finder tools (considered two advanced digital tools) that are not indicated as being used or implemented in the analysed SMEs. Further, the bivariate analysis regarding the undecided subjects about TmS raises an imposing future research question, namely, "Why were entrepreneurs responding differently to the question of using page speed loading analysis tools?". Could it be that they did not know that the Test My Site app is such a tool?

Proposing to measure the perceived usefulness of online presence and specific digital marketing tools by SME entrepreneurs, the authors draw on relevant results. Specifically, the low usage of trend analysis or tracking tools could threaten the creation and optimisation of digital marketing campaigns, and decision-makers may not even know it.

The authors expected to classify most SME entrepreneurs as Digital Achievers and Digital Investigators, but Digital Aspirants and Digitally Experienced are the most frequent types of entrepreneurs, and none of the subjects was identified as a Digital Investigator. However, it seems that most of them stated that they would like to know more about the tools they use. According to digital resilience scientific contribution [4,37,39,40], the authors propose Digital Experienced and Digital Aspirants as the most resilient typologies of entrepreneurs, while the least resilient is represented by the Untimely Divers, but the results required placing, in a special light, the entrepreneurs who need help to increase their skills to achieve entrepreneurial resilience. Given the revealed results, the entrepreneurial state of mind that SME entrepreneurs have is considered a barrier for the future development of their companies. If they do not accept help from specialists, studies published by Zighan et al., 2021 and Chen et al., 2021 [25,31] show that they can be completed with a self-learning perspective. In other words, it can be said that it is possible that SME entrepreneurs prefer to learn on their own how to develop the business with digital tools they already use.

Intending to focus on a strong discussion about this topic, the authors also desire to expose the weaknesses of the study in relation to the literature. As an important pillar [18], digital transformation cannot be presented as the single significant factor for SME development. In addition, the classification realised by authors can support the study conducted by [30], enriching the literature with digital marketing perspective, with 12 typologies of entrepreneurs but factors, such as gender, Social Media Marketing usage, and other relevant digital tools for management must be included for a big picture research. The respondents generated insights and valuable information about the economic significance of the businesses with long-term implications. Due to the evolution of technology deployed in digital transformation processes and the highly demanding need of consumers for

online presence, SMEs face unprecedented challenges. On one hand, the power of the entrepreneurs' openness to adapt to this context will represent a competitive advantage, even at the international level. On the other hand, improvement in the entrepreneurs' digital marketing skills presents multiple opportunities for business growth. The capacity of collaborating with specialists to promote goods and services online and the openness to learn new trends and applications can represent key factors for the survival of SMEs on the market. Moreover, the potential of developing the SMEs in an informational society aims to communicate and serve customers in a virtual space. In this regard, the mindset of the entrepreneurs to train the users to interact in an online environment can contribute to mitigation of the digital divide and to digital inclusion.

## 6. Conclusions

The researchers enrich the scientific literature in entrepreneurship with the perspective on the digital tools adopted and their perceived usefulness, analysing the behaviour of SME decision-makers. The study unveils the entrepreneurial state of mind of the SME owners to promote crystal clear actions for strategies adapted to high-demanding digital needs. The paper may be in the interest of both academia and business, with major implications for entrepreneurs.

The first limit is related to the probabilistic sampling method and the extrapolation should be viewed with caution due to the small size of the sample. Even so, the results obtained can be considered relevant for the studied topic and the chosen sample is a favourable factor for substantiating proposals for an improvement in the strategies implemented in such organisations. At the same time, the choice of topics for discussion and the selection of only certain applications to be researched can be a subjective perspective of researchers to address the topic. For this reason, a second limitation arises. Other limitations of the research are the lack of introduction of detailed characterization questions of the people behind the business and the study focuses more on the entities they represented.

Despite these limitations, the marketing research developed can provide a solid basis for conducting other research in the field, bringing value primarily to academia, because through it, one can learn about the behaviours, intentions, and perceptions of entrepreneurs, but new research opportunities in the fields of digital marketing and entrepreneurship can also be identified.

To the same extent, the managerial implications are strongly influenced by the research carried out because specialists in the field, companies, or business consultants could understand the challenges for SMEs and can create strategies with concrete actions to combat them.

A first future direction of research may be to conduct a study by introducing several questions in the support questionnaire to analyse the behaviour of entrepreneurs concerning the reactions of the target audience to the activity carried out in the online environment. Further, starting a new research study regarding the digital tools used by SME entrepreneurs by adding social media to the list as well may represent a new opportunity. Further, new extensive research aimed at studying the consumption behaviour of online service users can complete the results of the study. Moreover, a comparison can be made between SMEs from different countries, considering the DESI index and the Global Entrepreneurship Index, respectively, to see if there exist significant differences between them regarding the digitalization approach.

**Author Contributions:** Conceptualization, E.N., R.C.L., C.I.M., S.S., I.B.C., A.S.T. and G.B.; methodology, E.N., R.C.L. and G.B.; literature review, S.S., I.B.C. and A.S.T., analysis and writing the results, E.N., R.C.L. and C.I.M.; discussion and conclusions, E.N., R.C.L., C.I.M., S.S., I.B.C., A.S.T. and G.B.; writing—original draft preparation, E.N., C.I.M., S.S. and I.B.C., writing—review and editing, S.S.; visualization, A.S.T. and G.B.; supervision, G.B.; project administration, G.B.; funding acquisition, G.B. All authors have read and agreed to the published version of the manuscript.

**Funding:** This research was funded by the Transylvania University of Brasov.

**Informed Consent Statement:** Informed consent was obtained from all subjects involved in the study.

**Data Availability Statement:** Available on request from the corresponding author.

**Conflicts of Interest:** The authors declare no conflict of interest.

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
