# Peer review of "Unlocking the Entrepreneurial State of Mind for Digital Decade: SMEs and Digital Marketing"

_electronics, doi:10.3390/electronics11152358_

Round 1
Reviewer 1 Report
Digitalization is a growing phenomenon affecting the strategy, structure, and processes of many if not most of the businesses, with potentially large benefits for corporate performance. Digitalization is becoming largely embedded in products and services, and also increasingly in business processes. Despite these developments, there is limited empirical research on the state of digitalization and its implementation within companies. To my knowledge, the literature provides mixed evidence on the effects of digitalization on performance. Hence, there is need for any type of empirical study to grasp the diffusion of digitalization, including the survey-based one submitted here.
The submitted manuscript starts with a general introduction. In my view the introduction can be more precise by focusing carefully on how the results of the survey contribute to which literature. The analysis, however, is only loosely connected to the introduction due to a lack of clarity of underlying constructs and how the survey design attempts to capture the existence and relevance of these constructs. For instance, even the definition of the entrepreneur employed in the paper is an unusual one (p. 4: “the selection criteria for surveyed people were to be in charge and/or decision-making influence on spending a marketing budget allocated to developing an online presence in the organisation in which they are involved”) which might very well fit in some (salaried) managerial function rather than being an entrepreneur.
Some more specific issues are addressed below:
1. On the mechanics of the survey:
It is not clear how you get down to 377 responses if your total population is almost 450,000 SMEs.
It is also not clear how many potential participants have been conducted via which channel. So, a clear explanation of the sampling procedure is missing.
There is also no clear information / discussion of the many challenges faced by survey-based studies (importantly low response rates and selection biases).
2. Analysis:
It is not clear, why and how the Section 2.2. (“Analysis methods”) contribute to the paper. It looks to me as a collection of technical definitions without reference to the economic meaning and significance. I suggest, at a minimum, to explain this part better, or to remove it altogether.
The use of formulas in (7), (8), and (9) is redundant as these represent very basic definitions.
Equally unclear is the regression equation (10) where the dependent variable (OPI) is introduced after the equation is introduced. Eq. (10) explains OPI (online presence importance considered by SMEs entrepreneurs in various search engines). So, basically Eq. (10) is summary of what entrepreneurs think which search engine is useful to them. Perhaps I am missing something, but I do not really understand why the outcome of this exercise is going to be informative other than providing a set of casual observations.
I had difficulty to understand the basis of and the definition for creating the categories “Digital Aspirant, Digitally Experienced, and Digitally Informed”. The same applies to the remaining categories on page 12.
3. Relevance
The main messages that are listed in Section 4 (Discussion) are that entrepreneurs
-- consider their level of digital knowledge as intermediate,
-- are interested in specific digital tools to promote their business,
-- do not perceive the usefulness of working (for three months at least for special support) to enhance digital performance and do not intend to engage in such collaboration.
I am however, not convinced that these messages are of much importance, because there is no information on the economic significance of the businesses that the respondents conduct.
Author Response
Dear Reviewer 1,
Please find our response in the attached file.
Sincerely yours,
Authors

Reviewer 2 Report
The figures can be improve. Please add the value of axis in Figure 2. The Digital tools are hard to read in Figure 5. The is two Figures 5 (p. 12 and p. 13). The Figure 5 on page 13 is not clear, try to improve it (add the legend 'What does the circle mean?', for example). The text in Figure 7 is hard to read, too. There is a spelling error (in the third unexpected results 'Page speed ... oly 9.3% ...')
Author Response
Dear Reviewer 2,
Please find our response in the attached file.
Sincerely yours,
Authors

Reviewer 3 Report
Dear Authors,
The manuscript addresses an interesting question that links the SMEs and digital marketing. Based on survey methodology consisting of 333 entrepreneurs in Romania the study finds support for the perceived usefulness and need for more digital focus for SMEs.
The paper is exploring an interesting question, but in current state the study and methodology are at a preliminary stage.
1) Abstract: Abstract is too long winded and sentence structure is awkward. Authors should differentiate between the contribution of their study and the existing knowledge and focus more on the former in the abstract. Clearly state the problem to be addressed and the purpose of the research. Provide a concise summary that can stand-alone.
2) Introduction: In current form introduction is too long winded. Introduction is very general and lacked alignment with the research findings, theoretical and empirical existing literature. Manuscript should describe the problem to be addressed by the research, why is the problem important with rationale and justification including the consequences of not addressing it, and the corresponding purpose of the research.
3) Develop hypothesis based on literature and theory. For example, H0 and H1 are trivial and should not be part of the hypothesis section.
4) Summary statistics for various variables are missing. Manuscript should explain how the summary stats of important variables compare to the existing literature.
5) Discussion section needs to be reworded. In discussion of results explain the economic justification of results. Explain the results in light of similar papers on digital inclusion. Discussion needs to be a coherent and cohesive set of arguments that take the reader beyond this study and establish the findings in the existing literature. Contextualize the findings in the literature and explain the added value of your study towards that literature.
6) Figures and tables should be self-contained so that reader can understand them without going back to the text.
7) Careful English editing is required. I found the manuscript hard to read and construction of sentences awkward. Example:
8) Pg 4 “Also, the selection criteria for surveyed people were to be in charge and/or decision-making influence on spending a marketing budget allocated to developing an online presence in the organisation in which they are involved.”
Pg 16 “The first limit of the study is related to small number of entrepreneurs included which is the reason why the representativeness of the researched population cannot be ensured.”
Good luck with the manuscript!
Author Response
Dear Reviewer 3,
Please find our response in the attached file.
Sincerely yours,
Authors

Round 2
Reviewer 3 Report
Thanks for the revisions. Good luck with the next stage in the publication process.